# An Evaluation of the Effectiveness of Repetitive Transcranial Magnetic Stimulation (rTMS) for the Management of Treatment-Resistant Depression with Somatic Attributes: A Hospital-Based Study in Oman

**DOI:** 10.3390/brainsci13091289

**Published:** 2023-09-06

**Authors:** Intisar Al-Ruhaili, Salim Al-Huseini, Said Al-Kaabi, Sangeetha Mahadevan, Nasser Al-Sibani, Naser Al Balushi, M. Mazharul Islam, Sachin Jose, Gilda Kiani Mehr, Samir Al-Adawi

**Affiliations:** 1Psychiatry Residency Program, Oman Medical Specialty Board, Muscat 130, Oman; intisar.r18@resident.omsb.org; 2Department of Psychiatry, Al Masarra Hospital, Ministry of Health, Muscat 113, Oman; salimalhuseini@gmail.com (S.A.-H.); dr-saidkaabi@windowslive.com (S.A.-K.); 3Department of Behavioral Medicine, College of Medicine & Health Sciences, Sultan Qaboos University, Muscat 123, Oman; sm5520@nyu.edu (S.M.); naser.albalushi@hotmail.com (N.A.B.); 4Department of Statistics, College of Science, Sultan Qaboos University, Muscat 123, Oman; mislam@squ.edu.om; 5Studies and Research Section, Oman Medical Specialty Board, Muscat 130, Oman; sachin.j@omsb.org; 6Department of Neurology, Shariati Hospital, Tehran University of Medical Sciences, Tehran 14588-89694, Iran; gilda.kianimehr@gmail.com

**Keywords:** non-Western societies, depressive illness, somatic attributes, non-pharmacological approaches, repetitive transcranial magnetic stimulation (rTMS), treatment-resistant depression (TRD), Hamilton Rating Scale for Depression

## Abstract

Depressive illnesses in non-Western societies are often masked by somatic attributes that are sometimes impervious to pharmacological agents. This study explores the effectiveness of repetitive transcranial magnetic stimulation (rTMS) for people experiencing treatment-resistant depression (TRD) accompanied by physical symptoms. Data were obtained from a prospective study conducted among patients with TRD and some somatic manifestations who underwent 20 sessions of rTMS intervention from January to June 2020. The Hamilton Rating Scale for Depression (HAMD) was used for clinical evaluation. Data were analysed using descriptive and inferential techniques (multiple logistic regression) in SPSS. Among the 49 participants (mean age: 42.5 ± 13.3), there was a significant reduction in posttreatment HAMD scores compared to baseline (t = 10.819, *p* < 0.0001, and 95% CI = 8.574–12.488), indicating a clinical response. Approximately 37% of the patients responded to treatment, with higher response rates among men and those who remained in urban areas, had a history of alcohol use, and were subjected to the standard 10 HZ protocol. After adjusting for all extraneous variables, the rTMS protocol emerged as the only significant predictor of response to the rTMS intervention. To our knowledge, this is the first study to examine the effectiveness of rTMS in the treatment of somatic depression.

## 1. Introduction

Depressive disorders have been widely reported to be widespread around the world and one of the catalysts for dependency and disability. The World Health Organization (WHO) reports that depression is responsible for 4.3% of the worldwide burden of diseases and is predicted to become the third most common disease by 2030 [1]. Alarmingly, depression has become more widespread in the last decade; between 2005 and 2015, the number of individuals estimated to live with depression increased by 18.4% [2].

Depressive illnesses are amorphous forms of distress and are increasingly recognized to differ in their presentation from culture to culture [3,4]. The subject of depressive illness accompanied by somatic complaints in non-Western cultures has gained significant importance in the field of psychiatry [5]. Research has revealed that people from non-Western cultures tend to experience physical symptoms, such as pain or fatigue, as opposed to psychological symptoms while battling depressive illnesses. These physical symptoms, known as somatic complaints, are often viewed as an indication of a deeper underlying problem, such as a spiritual or social problem, in non-Western cultures. Psychological symptoms may not be well understood or accepted in these cultures, making people feel more comfortable discussing physical symptoms rather than psychological ones. Open expression of emotions, especially negative, can be viewed as a sign of weakness and can be perceived as disruptive to family and community harmony. In many cases, people may prefer to communicate their emotional distress through bodily complaints to avoid potential social stigma [6]. This cultural tendency to somaticize emotional distress can sometimes lead to difficulties in accurately diagnosing and treating mental health issues. Furthermore, stigma surrounding mental health problems can discourage people from openly discussing psychological struggles, leading them to present their distress in physical terms. Although the non-Western characteristics offer a case study, several characteristics can be generalised to the larger Arab world. Many Arab societies share similar cultural values and norms, emphasising collectivism, modesty, and respect for authority [7]. These shared values contribute to a cultural context in which somatic distress may be prevalent due to the desire to conform and avoid disrupting social harmony. As a result, depressive illness with somatic complaints in non-Western cultures may be underdiagnosed and undertreated [8]. Little research has been conducted to examine how somatic distress responds to emerging biomedical treatment protocols for depressive symptoms.

Under the biomedical model rubric, various psychotherapeutic modalities and pharmacological agents have been used to treat depressive symptoms [9,10]. In the field of biomedical care, brain stimulation therapies such as electroconvulsive therapy and repetitive transcranial magnetic stimulation (rTMS) often feature as some of the arrangements made to combat depressive illness [11]. Studies have shown that making lifestyle changes and using alternative therapies can be effective in reducing the symptoms of depression [12]. 

Among various approaches to mitigate depressive illness, there are widely recognised subgroups of people with depressive illness whose conditions are impervious to existing interventions [13]. Such clinical conditions sometimes fall under the umbrella of treatment-resistant depression (TRD). According to some estimates, approximately 30% of people with major depression do not respond to initial treatment and approximately 10–30% of those with major depression may experience TRD [14,15]. However, the exact prevalence of TRD is difficult to determine because there is no universally accepted definition of TRD, and the criteria used to define it can vary between studies.

rTMS is a relatively new method of managing TRD. rTMS is a non-invasive neuromodulation technique that has been used to treat depressive illnesses, particularly treatment-resistant depression (TRD) [16,17,18,19]. The exact mechanism of action of rTMS in the treatment of depression is not fully understood, but it is believed to work by modulating the neural circuits that underlie depression [20]. rTMS involves the application of magnetic pulses to the scalp, which generates an electrical current in the cortex. This current, in turn, leads to changes in the neural activity of the targeted brain regions and interconnected networks. The prefrontal cortex, which is involved in emotional regulation and decision making, is a common target of rTMS in the treatment of depression. By stimulating this area, rTMS can increase its activity and restore the balance of neural activity in the brain. This, in turn, can improve mood and reduce depressive symptoms. In addition to the prefrontal cortex, rTMS has also been used to improve the integrity of other regions of the brain that are involved in depression, such as the subgenual cingulate cortex and the amygdala. By modulating activity in these areas, rTMS can alleviate depressive symptoms and improve mood. In 2008, the US Food and Drug Administration (FDA) approved rTMS as a therapy for TRD. Compared to other neuromodulation techniques, such as electroconvulsive therapy, rTMS has several advantages, including a better safety profile, tolerability, and cost-effectiveness [21,22]. As a result, it has become a preferred form of therapy for TRD in many cases. rTMS has been reported as a monotherapy or an enhancement for TRD [23,24,25]. 

However, there is a significant dearth of data on rTMS from non-Western populations, and the Arabian Gulf is no exception. In a collective and traditional society such as Oman, explicit expression of emotion is often ostracised, and the task of improving life-related distress is often relegated to traditional healers [26]. Many of the maladaptive coping mechanisms developed by those in this region are associated with culturally specific idioms of distress [27]. Due to these factors, what is often labeled emotional distress is framed as ‘somatic metaphors’ or attributes. This presentation of distress is sometimes labeled as ‘atypical’ or ‘not otherwise specified’ (NOS), a classification used to describe a mental health diagnosis that does not fit any specific diagnostic category. It is used when the symptoms or conditions of a person are not severe or specific enough to meet the criteria for a particular diagnosis but are still significant enough to warrant clinical attention. Initial impressionistic observations have reported the usefulness of ad hoc electrical stimulation to mitigate presentations classified as NOS [28,29,30]. Little has been reported about the atypical presentation of depressive illness marked by somatic distress.

The present study aims to fill this gap in the existing literature by exploring the effectiveness of repetitive transcranial magnetic stimulation (rTMS) among service users in Oman who exhibited the core characteristics of major depressive disorders, were marked with somatic distress, and were impervious to pharmacological intervention.

## 2. Materials and Methods

### 2.1. Design and Setting of the Study

This is a naturalistic observational study that includes outpatients with treatment-resistant major depression who received rTMS treatment routinely between January and June 2020. The patients were referred to the Department of Neuromodulation and Sleep Disorders at Al-Massara Hospital in Muscat, Oman, to receive rTMS. Al-Massara Hospital is a tertiary care psychiatric hospital with full-fledged facilities in the neuromodulatory clinic, with referrals accepted from different regions of the country, as it is the only government hospital that provides this service.

### 2.2. Data Collection

Patients who underwent the rTMS intervention were evaluated for any conditions that could prevent safe use of rTMS. If any such condition was found, they were excluded from participating. Before starting treatment, informed consent was obtained as part of the standard procedure. The number of previous failed attempts with antidepressant drugs by patients was not restricted. They continued with their current drug combinations, without making any changes throughout the duration of the sessions (Table 1). Baseline assessments were conducted using the Hamilton Rating Scale for Depression (HAMD) just before the first session and after completing 20 sessions. The outcome measures consisted of the HAMD scores after the rTMS intervention, as well as the percentage of patients who showed a positive response to treatment.

### 2.3. Participant Inclusion and Exclusion Criteria

The participant pool included patients 18 years and older who had MDD according to the Composite International Diagnostic Interview (CIDI) and standardized diagnostic criteria such as the Diagnostic and Statistical Manual of Mental Disorders [31], as well as those who had the objective tendency to express distress in somatic language. The second inclusion criterion was that participants were required to have access to psychiatric services and must have received adequate treatment for depression, including at least two antidepressant trials in adequate doses and durations, and must not have shown a response or only a partial response to these treatments [32]. The third inclusion criterion was that individuals must have been experiencing persistent depression symptoms for at least 8–12 weeks despite adequate treatment and have a moderate to severe level of depression, as assessed by the standard measure, the Hamilton Rating Scale for Depression (HAMD). The final inclusion criterion was that the individual had to be willing to receive repetitive transcranial magnetic stimulation for the prescribed period.

Exclusion criteria for treatment-resistant depression included substance abuse disorders or a history of significant medical conditions, such as thyroid disorders, neurological disorders, or chronic pain, that could interfere with treatment. The study excluded patients who had a history of non-compliance with treatment, had received electroconvulsive therapy (ECT) in the last six months, or had received other supplements or novel treatments for depression in the preceding four weeks. Given that the study’s primary objective was somatic distress, participants whose scores were less than 13 on the somatic distress index, as assessed by the Bradford Somatic Inventory (which will be described below), were excluded from the study.

Ultimately, 49 patients were included in the study for the rTMS treatment intervention following the inclusion and exclusion criteria mentioned above. Details of the inclusion and exclusion criteria used in this study can be seen in Figure 1.

### 2.4. rTMS Protocol

Treatment was administered with a MagPro X100 stimulator equipped with the B70 fluid-cooled coil (MagVenture, Hovedstaden, Denmark). Two intervention protocols were used according to the doctor’s decision and the patient’s preference. The standard bilateral rTMS sequence consisted of a 1-Hz stimulation (120%) of the right dorsolateral prefrontal cortex (DLPFC) (120% of the resting motor threshold, 600 pulses per 10 min), followed by an FDA-approved (120% of the RMT, 3000 pulses: 4 s in, 26 s off per 37.5 min) 10-Hz stimulation (120%) of the left DLPFC.

Sequential bilateral TBS stimulation at the same stimulation sites and intensity (120% RMT) was administered to the right cTBS (50 Hz triple break pulse, 5 Hz for 600 pulses over 40 s) followed by the FDA-cleared left iTBS (50 Hz triple break pulse, 5 Hz, 2 s to activate, 8 s to disconnect, 600 pulses over 3 min and 9 s). Within the initial four treatments, an adaptive titration of 120% of the resting motor threshold (RMT) was implemented to improve tolerability. Two techniques, namely, beam F3 and the modified 5–6 cm rule, were used to locate the coil in the dorsolateral prefrontal cortex (DLPFC).

### 2.5. Outcome Measures

#### 2.5.1. Diagnosis of Major Depressive Disorders

All included subjects participated in a ‘gold standard interview’ using the style and format of the Composite International Diagnostic Interview (CIDI) [31]. The CIDI is a structured diagnostic interview designed to assess mental disorders according to the criteria of the *Diagnostic and Statistical Manual of Mental Disorders* (DSM) and the International Classification of Diseases (ICD). The CIDI was developed by the World Health Organisation (WHO) as a tool for use in large-scale epidemiological studies of mental disorders and has been used in many countries around the world. Semi-structured interviews were conducted without the interviewer’s knowledge of the results of other assessments.

#### 2.5.2. Somatic Attributes

The Bradford Somatic Inventory (BSI) is a 21-item questionnaire that taps into an individual’s tendency to express distress in the somatic language [33]. The BSI inquires about a wide range of somatic symptoms, such as back pain, headaches, dizziness, muscle pain, stomach pain, shoulder pain, etc. The BSI instrument has previously been used with the Omani population [34,35], with Arabic validation based on Zahid & Motaal [36]. Higher BSI scores suggest more preoccupation with somatic attributes. For this purpose, a cutoff score of 13 has previously been shown to yield a sensitivity rate of 72.5% and a specificity rate of 73.47% [36]. The purpose of this study was to detect somatic distress. To simplify the results, the composite scores were condensed into a ‘yes’ or ‘no’ category. However, the study did not consider variations in the somatic characteristics of BSI.

#### 2.5.3. The Hamilton Rating Scale for Depression

The Hamilton Rating Scale for Depression (HAMD) is used to quantify the variation and severity of depression symptoms. It is widely used to tap into depressive symptoms [37]. The HAMD comprises 17 items on Likert scales from 0 to 4 or from 0 to 2, with a total composite ranging from 0 to 61. From the corpus of literature, the cutoff scores range from 14 to 20, with 17 being the most common cutoff to differentiate those with depressive symptoms or otherwise [38]. Many interventional studies have used this cut-off point [39]. A 50% or more decrease in the HAMD score is often considered to indicate a positive response to treatment [39] (HAMD baseline−HAMD after treatment/HAMD baseline × 100%).

### 2.6. Data Analysis

After collection, data were revised, coded, tabulated, and analysed using IBM SPSS Statistics version 26. Descriptive variables were evaluated using central tendency measures, and differences between categorical and continuous variables were evaluated using the Chi-square test, *t* tests and ANOVA. Paired *t*-tests were applied to assess the changes in scale scores from baseline to the final session. A *p*-value less than 0.05 was considered statistically significant. Taking into account that the outcome variable was binary (response/no response), which was coded 1 to designate the response (if there was a reduction in the HAMD score < 50% of the baseline HAMD score) and 0 as otherwise, a multiple logistic regression model was used to identify potential predictors of the response to the rTMS intervention.

### 2.7. Ethical Approval

This study was carried out according to ethical guidelines, including the Declaration of Helsinki and the guidelines of the American Psychological Association on ethical human research [40]. The study was granted ethical approval by the Directorate General of Planning and Studies, Ministry of Health, Sultanate of Oman (MH/DGPS/DPT 740/2020). Measures were taken to ensure confidentiality, privacy, and appropriate management of study data.

## 3. Results

### 3.1. Profiles of the Participants

The details of the profiles of the respondents are shown in Table 2. In general, 49 participants completed 20 sessions of rTMS for the treatment of depression during the analysis period. Most of the participants were men (*n* = 34, 69.4%). The ages of the participants ranged from 22 to 82 years, and more than half (55.1%) were 40 years or older. The mean age of the patients was 42.5 ± 13.3 years, and most (71.4%) of the patients were living in urban areas. More than two-thirds (69%) of them were married. About 53.0% of them reported being employed. Most of the participants (81.6%) underwent the rTMS intervention for the first time and more than half (55%) of the participants had had rTMS on the left side of the brain. Of the 49 patients, 27 (55.1%) patients underwent the FDA standard 10 Hz protocol, and 22 (44.9%) patients received the theta-burst protocol (Table 2).

### 3.2. Evaluation of rTMS Treatment

A comparative analysis of the distribution of HAMD scores at baseline and follow-up (after 20 sessions of rTMS), as shown in Figure 2, revealed a striking feature. The distribution of HAMD scores at baseline was found to be highly positively skewed, with a mean score of 27.41 ± 7.46, while the distribution of HAMD scores at follow-up was found to be more or less symmetric, with a mean score of 16.88 ± 7.34 (Table 3). Figure 3 presents the individual and mean values of the HAMD at the beginning of the study and at the last follow-up observation. The results revealed a substantial decrease in HAMD scores, both at the individual and aggregate levels. There was a 38.4% decrease in mean HAMD scores: from 27.41 (range: 20–49, IQR = 13) at baseline to 16.88 (range: 5–39, IQR = 9) at follow-up (Figure 3). The reduction in the mean HAMD score after the rTMS intervention was found to be statistically significant (paired-sample test: t = 10.82, *p* < 0.001) (Table 3). The reduction in HAMD scores was also significant between the response and non-response groups, as well as between patients using the standard 10 Hz protocol and theta bursts (Table 2).

When evaluating the effectiveness of rTMS treatment among the 49 patients, it was observed that, in general, 18 (36.7%) showed a reduction of 50% or more in their baseline HAMD scores after 20 rTMS sessions, indicating a clinical response (Table 2). However, the response rate did not vary significantly between demographic variables (e.g., age, sex, occupation, marital status), place of residence, smoking and alcohol history at the residence, or clinical characteristics, except that the rTMS protocol used had some marginally significant (*p*-value < 0.10) effect on the response rate. About 48% of the participants who received 10 Hz rTMS had a response, compared to 23% of the patients who received the theta-burst protocol. Regarding stimulation laterality, in this study, we found that 48% of those who received rTMS in the left DLPFC showed a response to treatment, compared to those who received the bilateral rTMS protocol (23%). The response rate was also found to be higher among men, those who lived in an urban area, and those who had a history of alcohol consumption (Table 2).

To identify predictors of reduction in HAMD scores due to the rTMS intervention, multiple logistic regression analysis was used, the results of which are presented in Table 4. The results indicate the type of rTMS protocol used as the only significant predictor of reduction in the HAMD score after rTMS treatment. The FDA standard 10 HZ protocol was found to have more than five times higher odds of reducing the HAMD score after rTMS treatment compared to the theta-burst protocol (OR = 5.62, 95% CI = 1.022–30.94, *p* = 0.037).

## 4. Discussion

Treatment-resistant depression (TRD) refers to a type of depression that does not respond to standard antidepressant medications or psychotherapy. In cases of TRD, alternative treatments can be considered, such as a different type of antidepressant drug, psychotherapy, or a combination of both. Other treatments, such as ECT or TMS, may also be options for some people with treatment-resistant depression [41]. Although there is no unanimous definition of what constitutes TRD, it is generally agreed that the lack of remission despite two adequate trials of tolerable and evidence-based antidepressant treatment warrants the use of the term TRD [42]. According to the STAR * D study, approximately 63.2% of patients with TRD did not relapse after the first antidepressant drug trial, and approximately a third of the patients never achieved remission [41]. In the treatment of depression, lack of effectiveness is another challenge. A reduction of ≥50% in the severity of symptoms from baseline is the definition of effectiveness [43]. Antidepressant drugs have an effectiveness of up to 30% compared to placebo [44], but there is a dissenting view [13].

Given these data, the current management of TRD usually features different augmentation strategies. For example, the use of mood stabilisers and/or antipsychotics often results in polypharmacy, with potential interactions and associated safety concerns. To our knowledge, this is the first study in Oman that examines the effectiveness of rTMS among treatment-resistant depressive patients in the country. The therapeutic effectiveness of rTMS was examined in 49 outpatients with pharmacotherapy-resistant depressive symptoms.

The study involved 49 participants who underwent repetitive transcranial magnetic stimulation (rTMS) treatment for depression, which was quantified by structured and semi-structured interviews and marked with somatic complaints that were quantified using the Bradford Somatic Inventory. It should be noted that for all participants, this was their first time under rTMS. Most of the present cohort received the option of rTMS as part of their clinical care to improve what appeared to be treatment-resistant depressive symptoms. Importantly, it should be noted that demographic variables and intervention protocols did not show substantial variations between the different groups of participants, which increased the reliability of our findings. Most of the participants were male, lived in urban areas, and were married. The place of residence was significant here, as rTMS treatment requires regular follow-up. The tertiary centre where rTMS treatment was administered was located in an urban ‘area’.

The uniqueness of this study was in its approach that addressed the cultural nuances of expressing distress in Oman, a historically collectivistic culture in which emotions are not publicly expressed. Distress in Oman is often expressed through culture-specific idioms of distress, and emotional distress can be framed in somatic metaphors or attributed to physical symptoms. Somatic treatments such as TMS are likely to align with the cultural and social norms prevalent in Oman, where cultural patterning tends to consider the expression of emotional distress as taboo or even shameful. Somatic complaints can act as a culturally acceptable way of communicating distress without directly confronting the stigma associated with emotional distress. TMS, as a somatic intervention, aligns with this indirect mode of communication and may be more readily embraced by people who are uncomfortable with openly addressing psychological problems. Recognising and addressing these cultural differences, this study provides valuable information on the effectiveness of non-pharmacological interventions for depression in Oman. In psychiatric parlance, such presentations are likely to constitute an ‘atypical’ classification. The preponderance of physical symptoms is similar to depressive symptoms and is not limited to Oman. It is likely to be more common than previously acknowledged [8]. Patients with depressive symptoms associated with somatic complaints may be at risk of being underdiagnosed and undertreated [45]. As explained by Greden [46], physical symptoms such as pain, fatigue, and sleep disturbances are also common among depressed individuals belonging to European and US populations and can significantly affect their quality of life. However, these symptoms are often overlooked in clinical practise, leading to underdiagnosis and undertreatment of depression in these populations as well.

As the present data suggest, the present cohort included those whose condition did not improve in response to standard antidepressant medication. Although abundant studies have explored TRD, less attention has been paid to depressive illnesses with somatic characteristics. It is worth speculating on why the use of rTMS may be conducive in those societies where depression is often stigmatized. First, rTMS is a noninvasive and nonpharmacological treatment option, which may be more acceptable for people who are reluctant to seek biomedical care due to stigma or concerns about drug side effects. Second, repetitive transcranial magnetic stimulation (rTMS) is a physical treatment that is supported by explicit evidence and involves the use of a wire or helmet. This treatment complements the idiom of distress commonly observed in Omani patients, where emotions are not socially sanctioned. As a result, there is a prevalence of bodily manifestations. In the future, this sentinel study can be used as part of the evidence in favour of rTMS being a potentially valuable treatment option for people in societies where depression is stigmatised, providing an alternative to traditional treatment options that may be more acceptable or accessible to some people.

This study aimed to investigate potential qualitative differences between two modes of repetitive transcranial magnetic stimulation (rTMS) protocols: the 10 Hz protocol and theta-burst stimulation (TBS). In the existing literature, varying points of view have emerged. Some studies suggest that TBS could be a more time-efficient and cost-effective approach compared to conventional rTMS, while others propose that TBS might provide a more potent physiological form of stimulation than standard rTMS. Despite the contrasting outcomes reported in different studies, there seems to be a consensus that both theta-burst stimulation (TBS) and 10 Hz rTMS are effective treatments for depression. This study aligns with the prevailing perspective but diverges from previous research by indicating the effectiveness of both rTMS models in alleviating depressive symptoms, particularly those associated with somatic distress, as assessed by HAMD scores. This indirectly suggests that despite differences in the phenotypical presentation of somatic and non-somatic depression, rTMS holds promise for populations experiencing somatic distress.

Previous investigations have explored the impact of coil placement in the dorsolateral prefrontal cortex (DLPFC), specifically using the beam F3 technique and the modified 5–6 cm rule, revealing potential differences in rTMS effectiveness. The beam F3 technique has been lauded for its ability to accommodate individual head-size variations, resulting in a more precise and reliable targeting method within the DLPFC compared to the conventional 5–6 cm rule. The latter method entails functionally identifying the motor cortex and then moving the transcranial magnetic stimulation (TMS) coil a fixed distance anteriorly. However, the 5–6 cm rule is considered less reliable and has undergone adaptations in certain research contexts. However, the current study did not observe any significant distinction between the results of the beam F3 technique and the 5–6 cm rule for coil location within the dorsolateral prefrontal cortex. Nevertheless, it would be premature to conclusively state that there is no difference between the effects of 10 Hz and theta-burst stimulation or no disparities between the two coil localisation techniques for targeting the DLPFC. To comprehensively address these questions, randomised controlled trials are warranted. Some studies further enrich this line of thinking. Siddiqi et al. [47] have reported two specific neural circuits that are effective in addressing separate groups of depressive symptoms. Consequently, symptoms such as sadness and anhedonia (“dysphoria cluster”) tend to have a better response to one circuit, whereas anxiety and somatic symptoms (“anxiosomatic cluster”) tend to show better responses to stimulation of a different circuit. To support this view, studies suggest that different DLPFC locations targeted by rTMS may differentially modulate anxiosomatic and dysphoric symptom clusters derived from the HAMD [48]. However, more research is needed to fully understand the neural circuits underlying these symptom clusters and how best to target them with rTMS. If this line of research withstands further scrutiny and the purported ci, Mrcuits are further delineated, then the potential of addressing diverse sets of depressive symptoms will benefit more from different TMS treatments. This would lead to much-cherished personalised therapy [49].

Compared to current findings from previous studies, the response rate was found to be lower than that reported in the literature. For example, a study investigating the effectiveness of rTMS in clinical practise reported a response rate of 50% [50], which is higher than the present results or the previously significant set point for clinically significant effectiveness and acceptability [51]. Similarly, a multisite naturalistic study found a response rate of nearly 60% [52]. Furthermore, a meta-analysis included seven randomized controlled trials (RCTs) and evaluated the effectiveness of rTMS as a duration of the current depressive episode, and having a history of recurrent episodes rather than just one episode [21,53]. The present study found that 48% of patients who received 10 Hz rTMS experienced a 50% or greater reduction in the severity of their depressive symptoms compared to baseline. Importantly, we discovered a statistically significant difference in effectiveness between the 10 Hz rTMS group and the theta-burst group when considering other sociodemographic factors, a finding that is consistent with previous research [54].

Various protocols have been explored to treat TRD using rTMS, including unilateral rTMS targeting the left DLPFC, unilateral rTMS targeting the right DLPFC, and bilateral stimulation. A comprehensive review of these modalities consistently demonstrated their superiority over sham treatment [55]. In our study, nearly half of the participants (48%) who underwent rTMS targeting the left DLPFC experienced a 50% reduction in the severity of depressive symptoms. However, it should be noted increasing strategy in treatment-resistant depression (TRD) reported a response rate of 46% [56]. One trial indicated that certain factors predicted better response rates to rTMS. These factors included a higher intensity of stimulation and clinical characteristics such as less severe depression at baseline, but the difference observed between treatment groups did not reach statistical significance.

### Limitations

There are likely to be several limitations that confound the present study. The most obvious ones are highlighted here. First, the absence of a control group to compare the treatment or intervention group with a group that did not receive the treatment or intervention limits the generalizability of this study. Future studies should require more than a naturalistic observation, and a formal clinical trial is warranted in this regard, as the possibility of a placebo is significant [57]. Second, while concerted efforts were made to increase recruitment, the relatively small sample size of the present study means that the study may not have had enough statistical power to detect small but meaningful differences between groups. Third, participants in the present study were found to have somatic distress. This study was not equipped to examine the presence of side effects of antidepressants or withdrawal symptoms of antidepressants, which are known to be rife among patients receiving antidepressants. Therefore, it is not clear whether resistance to treatment was orthogonal to antidepressant side effects, antidepressant withdrawal symptoms, or somatic attributions.

In relation to this, it would have been ideal to chart whether the indices of somatic attributes improved in the 20 sessions of rTMS. However, the change in somatic distress was not addressed, as, to compare the existing literature, the Hamilton Rating Scale for Depression is the most widely used outcome measure. Finally, while the study indicated the effectiveness of rTMS, the study was not equipped to track whether there was sustained remission or otherwise at the follow-up. These limitations should be addressed in future studies.

## 5. Conclusions

Current natural observations among patients with resistance to treatment and the somatic idiom of distress that is simultaneously marked indicate the effectiveness of repetitive transcranial magnetic stimulation. Approximately 37% of participants with TRD responded to the intervention after completing 20 sessions, as quantified by the Hamilton Rating Scale for Depression. Although a large number of studies have documented a positive response rate to various versions of brain stimulation among people with TRD, this study explored its effectiveness among cross-cultural samples with documented somatic attributions. Previous studies have indicated that depressive disorders that manifest along with physical symptoms tend not to respond well to pharmacological or psychotherapeutic interventions. This is a sentinel study that requires further scrutiny using a more robust methodology in order to provide a more concrete solution to dealing with the enigma of TRD.

## Figures and Tables

**Figure 1 brainsci-13-01289-f001:**
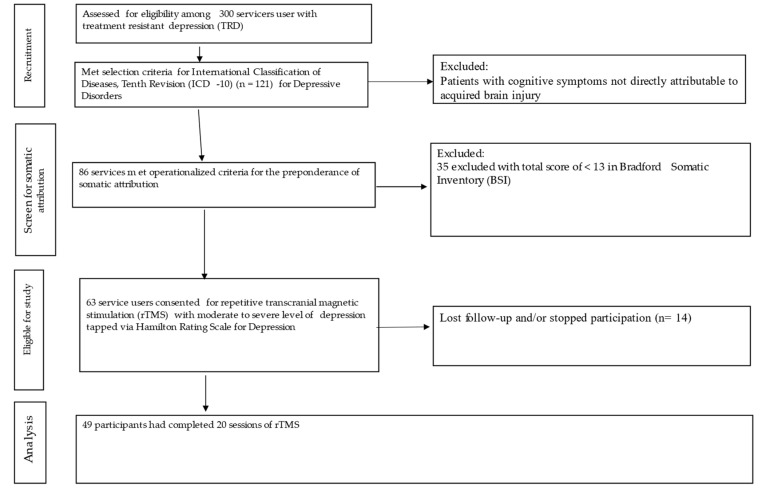
Flow diagram of study participants.

**Figure 2 brainsci-13-01289-f002:**
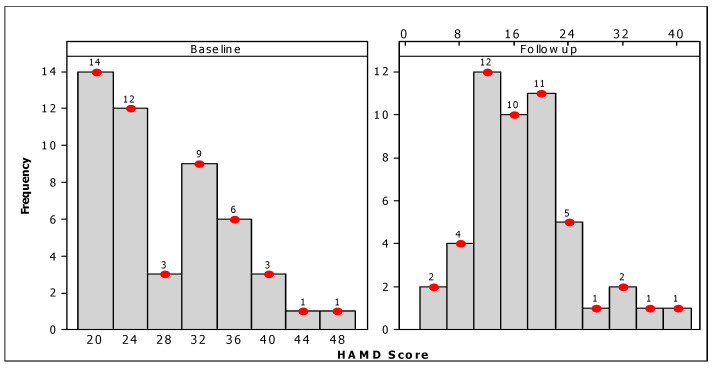
Distribution of baseline and follow-up outcome measures based on Hamilton Rating Scale for Depression (HAMD) scores.

**Figure 3 brainsci-13-01289-f003:**
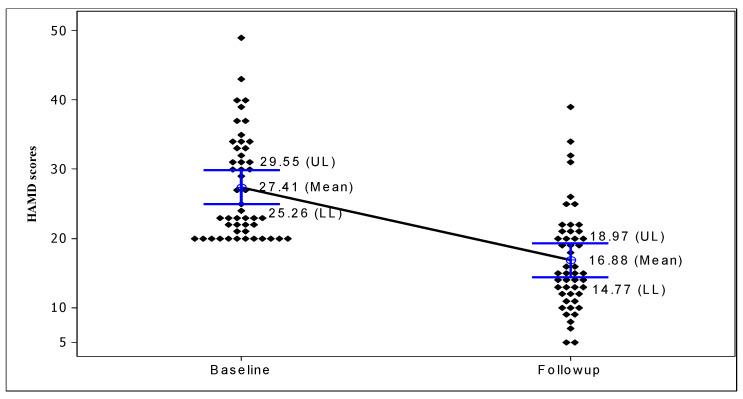
Change in individual and mean Hamilton Rating Scale for Depression (HAMD) scores after rTMS intervention with 95% confidence interval of mean scores.

**Table 1 brainsci-13-01289-t001:** Number of courses of antidepressant treatment.

Number of Courses of Antidepressant Treatment	*n*	%
Patients with 2 antidepressants as monotherapy	19	38.8
Patients with 3 antidepressants as monotherapy	20	40.8
Patients with 4 antidepressants	10	20.4

**Table 2 brainsci-13-01289-t002:** Frequency distribution of the patients according to sociodemographic and clinical characteristics and their corresponding responses to the rTMS intervention.

Variable	Total Sample *n* (%)	Reduction in the HAMD Score		
		No Response	Response	*p*-Value ^𝔛^
Total	49 (100)	31 (63.3)	18 (36.7)	
Gender				0.332
Male	34 (69.4)	20 (58.8)	14 (41.2)	
Female	15 (30.6)	11 (73.3)	4 (26.7)	
Age				0.519
<40	22 (44.9)	15 (68.2)	7 (31.8)	
40+	27 (55.1)	21 (57.1)	11 (42.9)	
Age, mean ± SD	42.53 ± 13.31	40.77 ± 12.89	45.56 ± 13.84	0.229
Place of Living				
Urban	35 (71.4)	21 (60.0)	14 (40.0)	0.453
Rural	14 (28.6)	10 (71.4)	4 (28.6)	
Marital status				
Single	15 (30.6)	10 (66.7)	5 (33.3)	0.743
Married	34 (69.4)	21 (61.8)	13 (38.2)	
Job				
Working	26 (53.1)	16 (61.5)	10 (38.5)	0.790
Not working	23 (46.9)	15 (65.2)	8 (34.8)	
History of alcohol				
Yes	6 (12.2)	3 (50.0)	3 (50.0)	0.259
No	24 (49.0)	17 (70.8)	7 (29.2)	
Unknown	19 (38.8)	11 (57.9)	8 (42.1)	
Previous rTMS				
Yes	9 (18.4)	5 (55.6)	4 (44.4)	0.595
No	40 (81.6)	26 (65.0)	14 (35.0)	
Target area				
Left DLPFC	27 (55.1)	14 (51.9)	13 (48.1)	0.066
Bilateral	22 (44.9)	17 (77.3)	5 (22.7)	
rTMS protocol used				
FDA standard 10 HZ	27 (55.1)	14 (51.9)	13 (48.1)	0.066
Theta bursts	22 (44.9)	17 (77.3)	5 (22.7)	

^𝔛^ *p*-values related to the chi-square test for categorical variables and *t*-test for quantitative variables. Note: Response rate defined as at least 50% reduction in the HAMD scale post-intervention.

**Table 3 brainsci-13-01289-t003:** Evaluation of the reduction in the Hamilton Rating Scale for Depression (HAMD) score after the rTMS intervention.

	Baseline Mean ± SD	Follow-Up Mean ± SD	t	*p*-Value
Overall	27.41 ± 7.46	16.88 ± 7.34	10.82	<0.001
Group				
Response	27.72 ± 7.57	11.17 ± 3.92	8.71	<0.001
Non-response	27.23 ± 7.46	20.19 ± 6.81	3.25	0.002
rTMS protocol used				
Standard 10 HZ	26.30 ± 6.02	14.04 ± 4.68	8.36	<0.001
Theta bursts	28.77 ± 8.87	20.36 ± 8.53	3.21	0.003

**Table 4 brainsci-13-01289-t004:** Logistic regression analysis of participant responses to the rTMS intervention.

	B	S.E. of B	Odds Ratio (OR)	95% CI for OR		*p*-Value
				Lower	Upper	
Age	0.020	0.034	1.020	0.955	1.090	0.554
Gender						
Male	0.113	0.943	1.120	0.176	7.109	0.904
Female (ref.)	0		1.000			
Place of Living						
Urban	0.741	0.836	2.098	0.407	10.803	0.376
Rural (ref.)	0		1.000			
Marital status						
Married	0.700	1.048	2.014	0.258	15.711	0.504
Single (ref.)	0		1.000			
Work Status						
Working	0.509	0.749	1.663	0.383	7.218	0.497
Not working (ref.)			1.000			
Previous rTMS						
Yes	0.362	0.934	1.436	0.230	8.957	0.699
No			1.000			
rTMS protocol used						
FDA standard 10 Hz	1.727	0.870	5.623	1.022	30.936	0.037
Theta-burst stimulation			1.000	1.000		
History of alcohol						
Yes	1.575	1.333	4.830	0.354	65.820	0.237
No			1.000			
Constant	−3.784	1.972	.023			0.055

## Data Availability

All data generated or analysed during this study are included in this published article.

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
