# Peer review of "An Evaluation of the Effectiveness of Repetitive Transcranial Magnetic Stimulation (rTMS) for the Management of Treatment-Resistant Depression with Somatic Attributes: A Hospital-Based Study in Oman"

_brainsci, 2023, doi:10.3390/brainsci13091289_

Round 1

Reviewer 1 Report

The article explores the effect of repetitive transcranial magnetic stimulation (rTMS) on treatment-resistant depression (TRD) with somatic distress in Oman. The study found a reduction in Hamilton Rating Scale for Depression (HAMD) score upon rTMS treatment. The research presents a promising opportunity to implement and expand the use of rTMS among TRD patients in Oman, where sociocultural pressures may impede the adoption of alternative therapies. The novelty of the study lies in the unique execution setting rather than the treatment itself. To enhance the article's suitability for publication, certain key points should be addressed:

  1. In the context of what is present in line 139, please include a table that shows the number of antidepressant treatments that the patients underwent.

  2. In line 275, please correct the typo in HAMD score to 27.41. 

  3. In figure 3, please omit the Gaussian fit. As the data is quite skewed, especially for baseline, and does not approach the central limit, the Gaussian fit looks inconsistent. 

The standalone figure at the start of page 9 should suffice for this.

  1. Also, please include a connecting plot, showing the HAMD data from baseline and follow up for individual patients.

  2. Please provide an evaluation of somatic distress within the Arab/Middle Eastern context in general, and subsequently highlight Oman's unique characteristics in this regard. Alternatively, you may present the case of Oman and then identify features that can be generalized to the broader Arab world.

  3. The rationale for why rTMS would be a more acceptable treatment option in Oman's social context requires further elaboration and strengthening.

  4. The absence of controls is a significant limitation of the study, especially when considering its presentation within a distinctive social setting. Though acknowledged as a limitation, it is crucial to include this aspect in the paper to draw credible conclusions.

Addressing these points will substantially enhance the article's quality.

Author Response

Please see the authors' response to the reviewer 1 comments

Reviewer 2 Report

The authors wanted to examine the real-world effectiveness of 20 sessions of the rTMS intervention for treating patients with TRD and some somatic manifestations. I only have a few comments to further improve the quality of the authors’ paper. I have outlined these issues below:

Materials and methods

1. It is not clear this study was a real-world, naturalistic observal study or a prospective open-label trial.  

2. Functionality scale was lacking.

3. efficacy vs effectiveness

I thought this study seemed to examine the real-world effectiveness of rtms in standard medical practice in TRD patients somatic manifestations by using a study design of a prospective, observational (non‐interventional), cohort study.

4. exclusion criteria

Line 161

included comorbid psychiatric disorders, such as bipolar disorder, schizophrenia, or substance abuse disorder. 

I suppose bipolar disorder or schizophrenia should not be comorbid psychiatric disorders for a patient with major depressive disorder. 

5. 

Line 95

rTMS has also been used to stimulate other regions of  the brain involved in depression, such as the subgenual cingulate cortex and the amygdala

I suppose rTMS is unable to directly stimulate sgACC and amygdala.  

6.

In the flow diagram

     35 participatns with BSI total score < 13 were excluded.  This was not mentioned in the exclusion criteria.

7.

Line 275

the mean HAMD scores: from 2.7.41 (range: 20-49, IQR=13) at baseline to 16.88 

a typo  27.41

8.

Table 2  t values for response and non-response seemed to be misplaced.

Discussion

9.

The authors discussed their results in a relatively superficial way. Let's say in that from line 342

Most of the patients had received rTMS for the first time and more than half of them had rTMS on the left side of the brain. The study found that 48% of the participants who received the FDA standard 10 Hz rTMS protocol had at least a 50% reduction in the severity of the symptoms, compared to only 23% in the Theta burst group. Participants who received rTMS in the left DLPFC showed a better response to treatment than those who received bilateral rTMS. About 37% of the participants showed a significant improvement in their scores on the HAMD assessment. Demographic variables and intervention protocols did not differ significantly between groups. 

The statement ovelaps with results. 

10.

The authors may have to stress that treatment-resistant depression with somatic attributes was the target population in this study. 

Therefore, the authors may have to provide the evaluation of the reduction in the BSI total score after the rTMS intervention in the results and discuss them in the discussion.

11.

The authors focused on TRD patients with somatic attributes. They also found the difference in effectiveness between the 10 Hz rTMS versus the Theta burst stimulation. This would be a focus to be discussed. 

In addition, they had two techniques, namely beam F3 and the modified 5-6 cm rule, were used to locate the coil in the dorsolateral prefrontal cortex (DLPFC). That factor was also a focus to be shown in the results and to be discussed. 

 Please see the following reference where the neural circuits correlated with anxiosomatic culster and dysphoria culster dervied from HAMD may be differentially modulated by different DLPFC locations targeted by rTMS.  

Siddiqi SH, Taylor SF, Cooke D, Pascual-Leone A, George MS, Fox MD. Distinct Symptom-Specific Treatment Targets for Circuit-Based Neuromodulation. Am J Psychiatry. 2020 May 1;177(5):435-446. doi: 10.1176/appi.ajp.2019.19090915. Epub 2020 Mar 12. PMID: 32160765; PMCID: PMC8396109.

12. 

In HAMD scale,  there are 6 items combined together to represent anxiety- somatization (AS) factor   

item 10 (psychic anxiety),      0-4

item 11 (somatic anxiety),   0-4

item 12 (somatic symptoms-gastrointestinal),   0-4 

item 13 (somatic symptoms-general),    0-2

item 15 (hypochondriasis),     0-4

item 17 (insight).    0-2

HDRS AS score    Total  0-20

The readers may wonder the correlation between the participants'  BSI total score and HDRS AS score.  

In the reviewer’s opinion, the above-mentioned issues need to be addressed by the authors.

Author Response

Please see the enclosed response to the reviewer 2 comments

Round 2

Reviewer 2 Report

In addition to a minority of comments I outline below, the authors have answered all of my previous comments and have revised the manuscript accordingly.

1.

Line 161 included comorbid psychiatric disorders, such as bipolar disorder, schizophrenia, or substance abuse disorder.

The authors suggested to refer to a Delphi-method-based consensus guideline for definition of treatment-resistant depression for clinical trials (Mol Psychiatry . 2022 Mar;27(3):1286-1299. doi: 10.1038/s41380-021-01381-x. Epub 2021 Dec 15).

However, in the review article, schizophrenia was not clearly mentioned. Most importantly, in the consensus about "Clinical presentation", it was stated that "

Comorbid personality disorders or other mental disorders should be excluded from TRD/PRD studies only when their onset is properly documented as independent and antecedent to the MDD diagnosis."

In DSM-5, the criteria for schizophrenia item  D. Schizoaffective and Mood Disorder exclusion  "Schizoaffective disorder and depressive or bipolar disorder with psychotic features"  have been ruled out…”

Thus, it is unlikely that a patient diagnosed with MDD has comorbid schizophrenia.

2.

Line 391

As explained by Greden [46],

The body size of reference 46 is too big.

Author Response

Our responses to the reviewer 2 comments are detailed in the attached file using a point-counterpoint formal
